# Text-Guided Video Amodal Completion

## Abstract

Amodal perception enables humans to perceive entire objects even when parts are occluded, a remarkable cognitive skill that artificial intelligence struggles to replicate. While substantial advancements have been made in image amodal completion, video amodal completion remains underexplored despite its high potential for real-world applications in video editing and analysis. In response, we propose a video amodal completion framework to explore this potential direction. Our contributions include (i) a synthetic dataset for video amodal completion with text description for the object of interest. The dataset captures a variety of object types, textures, motions, and scenarios to support zero-shot transferring on natural videos. (ii) A diffusion-based text-guided video amodal completion framework enhanced with a motion continuity module to ensure temporal consistency across frames. (iii) Zero-shot inference for long video, inspired by temporal diffusion techniques to effectively manage long video sequences while improving inference accuracy and maintaining coherent amodal completions. Experimental results shows the efficacy of our approach in handling video amodal completion, opening potential capabilities for advanced video editing and analysis with amodal completion.

## 1 Introduction

Humans possess an extraordinary ability to perceive objects as complete entities, even when parts are obscured. In everyday life, objects frequently block one another from view. Yet, we effortlessly identify and reconstruct their hidden portions—a capability known as amodal perception or amodal completion (Kellman & Shipley, 1991; van Lier, 1999). Human's visual system achieves this by relying on shape continuity, symmetry (van Lier, 1999), and a deep familiarity with the world around us (Yun et al., 2018). Replicating this cognitive process of amodal completion presents a significant challenge for artificial intelligence (AI), despite recent advances in computer vision. Achieving amodal completion in AI could benefit diverse applications, including robotics (Qin et al., 2020; Back et al., 2022), autonomous driving (Qi et al., 2019), and augmented reality (Ozguroglu et al., 2024). Similar to other human visual abilities, amodal perception has inspired the development of AI algorithms designed to mimic this capability. Research has initially focused on amodal segmentation (Li & Malik, 2016; Follmann et al., 2018; 2019; Tran et al., 2022; Yao et al., 2022; Gao et al., 2023), where models attempt to obtain the object's complete shape. More recently, the advent of denoising diffusion models (Ho et al., 2020; Rombach et al., 2022) has spurred progress in amodal content completion (Ozguroglu et al., 2024; Xu et al., 2024; Zhan et al., 2024).

Despite significant advancements in image amodal completion (Ozguroglu et al., 2024; Xu et al., 2024; Zhan et al., 2024), research in video amodal completion remain unexplored. This is primarily due to the challenges posed by temporal dimension and dynamic occlusions in video data, despite the broad applications in fields like robotics (Back et al., 2022), autonomous driving (Qi et al., 2019), video editing (Ling et al., 2020), and content creation (Chu et al., 2024). Unlike static images, video requires models to simultaneously track and complete occluded objects consistently across frames, enabling seamless continuity and realistic rendering of occluded regions. Additionally, a key limitation of prior image amodal completion datasets is their lack of auxiliary information to describe occluded content (Fan et al., 2023). This absence is particularly problematic in cases of significant occlusion, where inferring and completing hidden parts becomes ambiguous or ill-posed. To address these challenges, we present three key contributions in this work as follows:

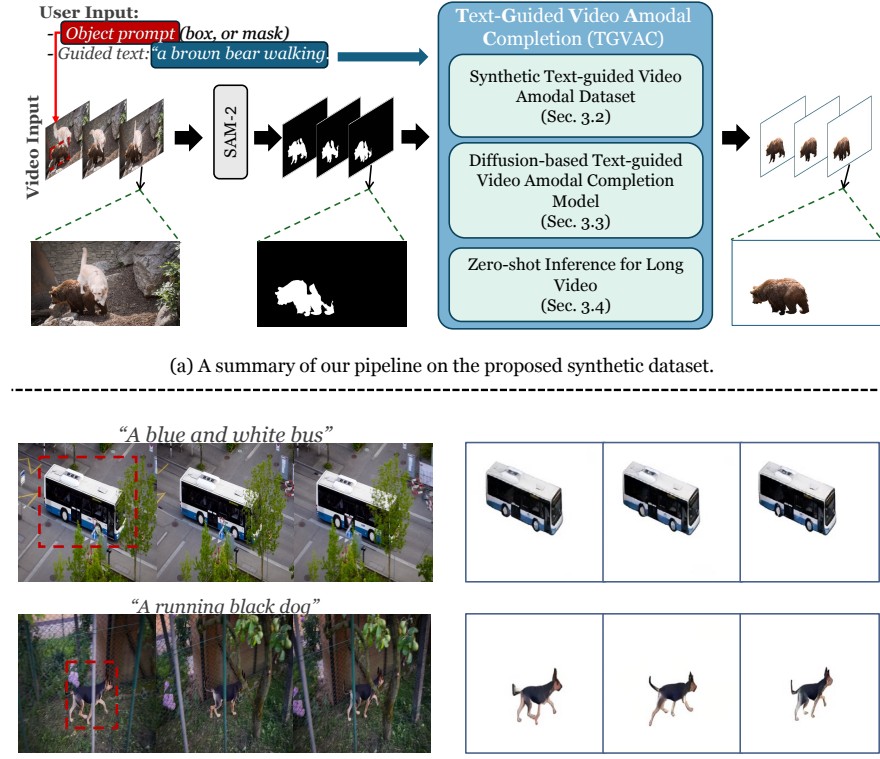

(a) A summary of our pipeline on the proposed synthetic dataset.

(b) Zero-shot transfer results on natural videos using TGVAC.

Figure 1: An overview of our text-guided video amodal completion pipeline. Given an input video, users select an object of interest in the first frame and provide a text description of the expected output. Our pipeline then generates a completed video, filling in the missing shape and texture of the object. (Top) A summary of our proposed pipeline and three key contributions. (Bottom) Zero-shot transfer results on natural videos using our method.

*i. Synthetic Text-guided Video Amodal Completion Dataset*: We introduce a synthetic video amodal completion dataset. This dataset encompasses diverse object categories and scenarios, providing knowledge on various shapes, textures, and motions to facilitate zero-shot transferring. To address the inherent ambiguity in amodal completion, we enhance the dataset with detailed textual descriptions of occluded regions, offering explicit guidance for accurately completing hidden content.

*ii. Diffusion-based Text-guided Video Amodal Completion Model:* We propose a novel framework for video amodal completion that generates complete object shapes, textures, and motions. As illustrated in Figure 1, given an input video, our proposed text-guided video amodal completion aims to extract the object and fill in occluded areas with semantically coherent information. To the best of our knowledge, this is the first exploration of amodal completion in videos. Inspired by the recent advancements video generation (Ho et al., 2022a; Guo et al., 2024; Blattmann et al., 2023), we leverage diffusion models to establish our baseline. Our approach employs a two-phase training strategy: frame-level training and motion training. In the first phase, we train a denoising UNet at the frame level to effectively capture spatial features. In the second phase, we focus on training motion layers while keeping the frame-level layers frozen, ensuring temporal coherence across frames.

*iii. Zero-shot Inference for Long Video:* Inspired by MultiDiffusion (Bar-Tal et al., 2023), Temporal Diffusion (Zhang et al., 2024), our approach manages long videos by dividing them into training-sized clips. The resulting completions are then integrated to ensure consistency across the entire video.

Experimental results show that our framework effectively outperforms the existing frame-level amodal completion methods as well as show the capability of zero shot-transferring to natural videos, unlocking new possibilities for advanced video editing and analysis through video amodal completion.

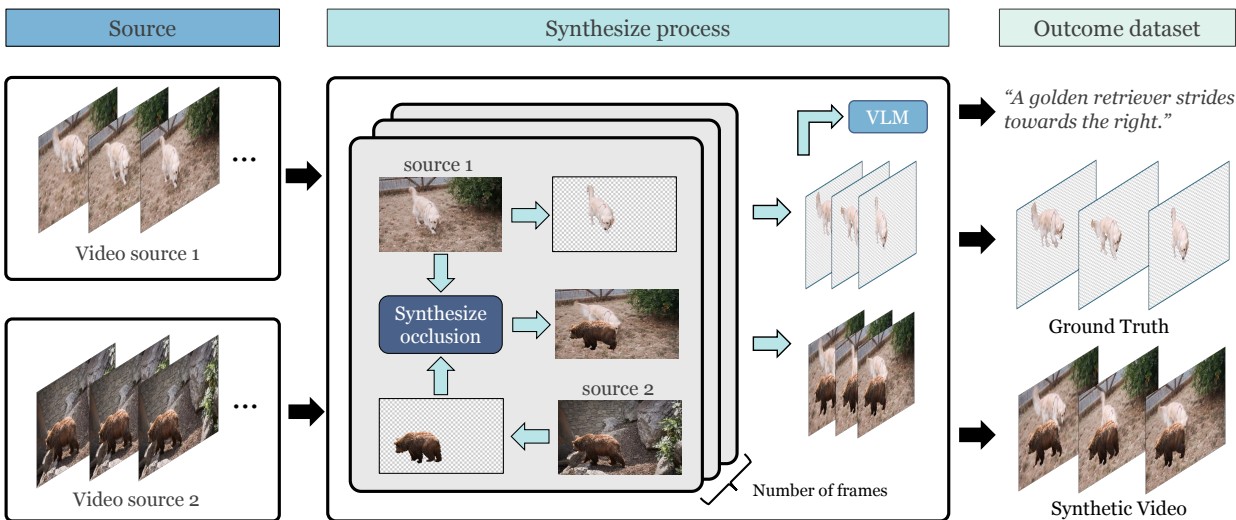

Figure 2: **Synthesizing training data process.** We selected videos with unoccluded objects and used provided masks to isolate object pixels. These were used to synthesize occlusion videos and create corresponding amodal completion ground truth. A vision language model (e.g. BLIP (Li et al., 2023) is utilized to generate ground truth's text description.)

## 2 Related Work

### 2.1 Amodal Completion and Segmentation

Amodal visual understanding has been explored in various aspects to complement normal visual understanding, often resulting in outputs obscured by foreground objects. For example, amodal segmentation generates a complete mask of a particular object (Zhu et al., 2017; Ke et al., 2021; Qi et al., 2019; Reddy et al., 2022; Ling et al., 2020). Amodal detection predicts entire objects, including hidden parts (Kar et al., 2015; Hsieh et al., 2023). More recently, amodal completion aims to generate the complete shape of an object (Ehsani et al., 2018; Zhan et al., 2020; Ozguroglu et al., 2024). The first two tasks have been well-explored, mainly due to advancements in model domain methods for visible mask problems. Furthermore, thanks to numerous closed-world datasets (Zhan et al., 2020; Ling et al., 2020; Ke et al., 2021; Qi et al., 2019; Kar et al., 2015; Zhu et al., 2017) and large synthetic datasets (Ehsani et al., 2018), amodal segmentation methods have developed significantly, such as PCNet (Zhan et al., 2020) or AISFormer (Tran et al., 2022). Controlling and generating whole objects is a more challenging task due to the non-trivial nature of conditioning on visible masks for generation. Pix2gestalt (Ozguroglu et al., 2024) addresses this challenge preliminarily by fine-tuning a large-scale diffusion model on a synthetic dataset. Despite significant advancements in image amodal completion (Ozguroglu et al., 2024; Xu et al., 2024; Zhan et al., 2024), there has been limited exploration of video amodal completion. In this work, we aim to bridge this gap by studying the challenges and opportunities within the video amodal completion problem.

### 2.2 Diffusion Models

Denoising Diffusion Probabilistic Model (Ho et al., 2020), or DDPM, has emerged as one of the most powerful methods among generative models. It is widely applied in various computer vision tasks, including image (Ho & Salimans, 2022; Song et al., 2021; Dhariwal & Nichol, 2021) and video generation (Blattmann et al., 2023; Ho et al., 2022a; Gu et al., 2023), motion generation (Guo et al., 2024; Zhang et al., 2024), 3D generation (Liu et al., 2023; Deitke et al., 2023; Wu et al., 2024), and out of computer vision like waveform generation (Kong et al., 2021). The breakthrough starts with (Dhariwal & Nichol, 2021) demonstrating that diffusion models can outperform GANs (Goodfellow et al., 2014) in image synthesis. This is followed by the success of Stable Diffusion (Rombach et al., 2022), trained on LAION-5B (Schuhmann et al., 2022), which

offers improved computational efficiency. One of the key features of Stable Diffusion is that we can guide it using text prompt, which leads to many advances in image editing (Brooks et al., 2023; Gal et al., 2022; Ruiz et al., 2023). Recently, diffusion models also have been explored in video syntheis. Video Diffusion Models (VDM) (Ho et al., 2022b) naturally extend the concept of text-to-image diffusion models by training on both image and video datasets. Imagen Video (Ho et al., 2022a) creates a cascade of video diffusion models and incorporates spatial and temporal super-resolution techniques to produce high-resolution, time-consistent videos. Make-A-Video (Singer et al., 2022) builds on text-to-image synthesis models and employs unsupervised video data to enhance performance. Gen-1 (Esser et al., 2023) expands on SD by introducing a structure- and content-guided video editing approach, using either visual or textual descriptions of the desired outcome. Tune-A-Video (Wu et al., 2023) introduces a novel task of one-shot video generation, extending SD with a single reference video.

## 3 Text-Guided Video Amodal Completion

Given an RGB video $\mathbf{v} = \{\mathbf{x}_i\}_{i=0}^N$, an initial prompt $\mathbf{p}_0$ (e.g., a bounding box or segmentation mask) specifying an object of interest $\mathbf{o}$ in the first frame $\mathbf{x}_0$, and a text description $y$ of that object, the amodal video completion task involves identifying both the visible and occluded parts of $\mathbf{o}$ throughout the video sequence $\mathbf{v}$. In practice, users typically indicate the object (for example, by drawing a bounding box) and provide a textual description. Traditional amodal completion relies solely on the object prompt, but when the object is largely occluded, inferring its hidden parts can be challenging (Xu et al., 2024). By incorporating the text description $y$ into our setting, we leverage the capabilities of pretrained text-to-image diffusion models, which can generate images based on textual prompts, thereby improving the overall amodal completion.

Formally, let $\mathbf{v}_{\text{out}}$ denote the output video depicting the complete object of interest $\mathbf{o}$, we have:

$$\mathbf{v}_{\text{out}} = f_\theta(\mathbf{v}, \mathbf{p}_0, y) \tag{1}$$

Here, $f_\theta(\cdot)$ denotes an estimator function, such as a conditional diffusion model. The goal is for the visible portions of the object $\mathbf{o}$ in $\mathbf{v}_{\text{out}}$ to accurately match the corresponding visible mask of $\mathbf{o}$ in the input video $\mathbf{v}$. Additionally, the completed (occluded) portions should integrate seamlessly, maintaining contextual consistency and avoiding any physically implausible object configurations.

### 3.1 Preliminaries

Diffusion models (Ho et al., 2020) aim to learn the data distribution $p(\mathbf{x})$ by sequentially denoising images. In the forward process, noise is gradually added to an image $\mathbf{x}$ over $T$ time steps, transforming it into a sample with nearly Gaussian noise. In the reverse process, the model learns to remove this noise over $T$ steps. At each step $t = [1, T]$, a neural network predicts the noise $\epsilon_\theta(\mathbf{x}^t, t)$ for the noisy image $\mathbf{x}^t$. Unlike standard diffusion models that operate directly on image pixels, latent diffusion models (LDMs) (Rombach et al., 2022) work in the latent space of pre-trained autoencoders. Given an image $\mathbf{x} \in \mathbb{R}^{H \times W \times 3}$, an encoder $\mathcal{E}$ encodes $\mathbf{x}$ into a latent representation $\mathbf{z} = \mathcal{E}(\mathbf{x})$, and a decoder $\mathcal{D}$ reconstructs $\mathbf{x}$ from $\mathbf{z}$ as $\hat{\mathbf{x}} = \mathcal{D}(\mathbf{z})$. In this framework, the autoencoder functions as a time-conditional UNet (Ronneberger et al., 2015), denoted as $\epsilon_\theta(\mathbf{z}_t, t)$, where $t$ is a specific time step and $\mathbf{z}_t$ is the added noise latent representation at that step. To incorporate an input condition $y$, such as images, masks, text, LDMs integrate cross-attention layers (Vaswani et al., 2017) into the denoising UNet, enabling $y$ to map to the intermediate layers of the UNet (Rombach et al., 2022).

### 3.2 Synthetic Text-Guided Video Amodal Completion Dataset

A significant obstacle in amodal completion research is the scarcity of natural image datasets that include ground truth for amodal scenarios. Previous studies on image amodal completion (Ozguroglu et al., 2024; Xu et al., 2024; Zhan et al., 2020) address this by generating pseudo-occluded images. However, amodal completion dataset for video-level have been unexplored. To tackle the challenge, we develop a synthetic dataset tailored for this specific purpose. We aim to ensure our dataset encompassed a wide range of common

Table 1: High level statistic of our synthetic dataset regarding each source.

| | **DAVIS** Pont-Tuset et al. (2017) | **YTVOS** Xu et al. (2018) | **LaSOT** Fan et al. (2019) |
|---|---|---|---|
| Mean #frames | 57 | 113 | 1,797 |
| # object classes | 9 | 65 | 60 |
| # instances | 24 | 332 | 1,044 |
| # synthesized video | 120 | 1,600 | 5,220 |
| # average object size | 12.80% | 13.61% | 10.31% |
| # average occlusion rate | 0.29 | 0.39 | 0.35 |
| # average occlusion-coverage rate | 0.40 | 0.37 | 0.42 |

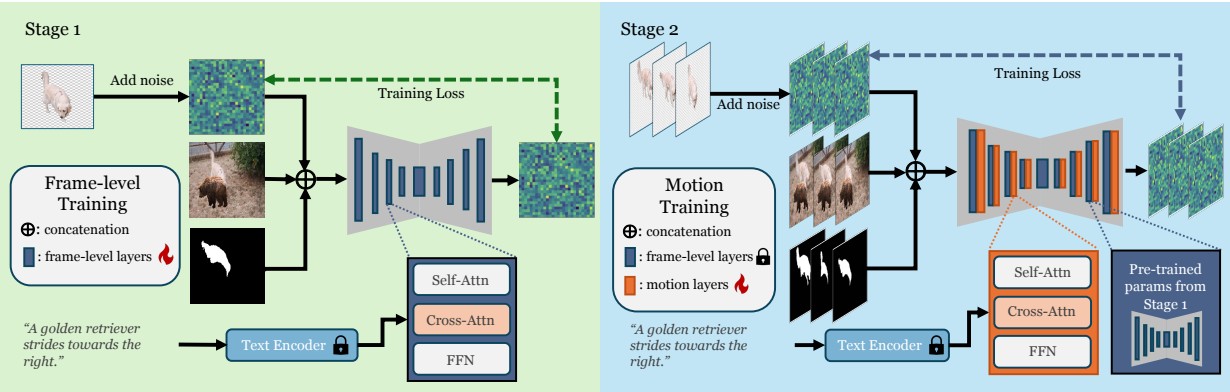

Figure 3: **Training pipeline of the proposed method.** Our approach follows a common two-stage training strategy: first, training a denoising UNet at the frame level to capture spatial features, and then incorporating motion training to ensure temporal coherence.

objects and scenarios, enabling the extraction of knowledge related to various shapes, textures, and motions to support zero-shot transferring ability. Previous amodal completion studies (Ozguroglu et al., 2024; Xu et al., 2024; Zhan et al., 2020) typically create pairs of images where a single, whole (unoccluded) object is overlaid with an occluder, producing a pseudo-occluded image as input and its corresponding whole counterpart as ground truth. We adopted and expanded upon these strategies to create a comprehensive video amodal completion dataset, resulting pairs of pseudo-occluded videos as inputs and their corresponding videos of single whole objects as ground truth. However, a notable limitation of prior image amodal completion datasets is the lack of auxiliary information to describe the occluded object, particularly in cases of significant occlusion, making it challenging to infer the hidden parts(Xu et al., 2024). To address this, we enhanced our dataset by including a textual description for each video data point, which describe the single whole object in the ground truth and serves as an extra conditional input. In specific, we sourced videos from video object segmentation datasets such as DAVIS (Pont-Tuset et al., 2017), YTVOS (Xu et al., 2018), and LaSOT (Fan et al., 2019). These datasets were chosen because their videos typically feature a single object of interest with minimal occlusion and significant motion variation, making them suitable as ground truth for amodal tasks. For DAVIS (Pont-Tuset et al., 2017) and YTVOS (Xu et al., 2018), we first selected videos featuring objects without occlusion and used the provided masks throughout the video frames to isolate the object pixels. These were then used to create synthesized occlusion videos and corresponding amodal completion ground truth. As illustrated in Figure 2, given videos of a dog walking and a camel walking, we utilized their annotated masks to create occlusion scenarios by overlapping them, generating one occlusion video as input and retaining the complete, unoccluded pixels as ground truth. The LaSOT (Fan et al., 2019) dataset, originally a video object tracking dataset lacking annotated masks, was incorporated due to its extensive variety in object types, motions, and large video count, with most videos containing a single object. We leveraged SAM (Kirillov et al., 2023) to obtain initial segmentation masks using the annotated bounding boxes and subsequently refined these masks to correct any inaccuracies. For the textual description, we leverage BLIP-2 (Li et al., 2023) to generate the textual caption given the first frame of the ground truth video.

Table 1 provides detailed statistics of the dataset, breaking down the object classes, number of instances, total synthesized videos, average object size, average occlusion rate, and average occlusion-coverage rate for each source. Regarding the occlusion-coverage rate, particularly, we quantify the spatial spread of an occluder over the entire video over time. For each frame, $i \in \{1, \ldots, N\}$, the binary intersection $I_i = A_i \cap O_i$ between the main-object amodal mask $A_i$ and the occluder mask $O_i$; we then form the per-video unions $U_{\text{occ}} = \bigcup_{i=1}^{N} I_i$ and $U_{\text{obj}} = \bigcup_{i=1}^{N} A_i$. Then the occlusion-coverage ratio for each video is computed as: $\frac{|U_{\text{occ}}|}{|U_{\text{obj}}|}$, where $|\cdot|$ denotes the cardinality (number of foreground pixels). An occlusion-coverage ratio of 0 indicates that the occluder never overlaps the object, while an occlusion-coverage ratio of 1 means that every object pixel is hidden at least once during the sequence. This metric offers an indication of how much the occluder interacts with the object spatially and temporally throughout the sequence.

### 3.3 Diffusion-Based Text-Guided Video Amodal Completion

Given the problem definition in 1, with an RGB video $\mathbf{v} = \{\mathbf{x}_i\}_{i=0}^{N}$ and an initial prompt $\mathbf{p}_0$ (e.g. points, bounding box, or mask) specifying an object of interest $\mathbf{o}$ in the first frame $\mathbf{x}_0$, we utilize SAM-2 (Ravi et al., 2025) to obtain the visible masks $\mathbf{v}_m = \{\mathbf{m}_i\}_{i=0}^{N}$ of $\mathbf{o}$ across all frames where $\mathbf{m}_i$ is the visible mask of $\mathbf{o}$ at frame $i$.

To the best of our knowledge, there are no existing works that have explored amodal completion in video data. In this study, we propose a baseline method for this task, named **T**ext-**G**uided **V**ideo **A**modal **C**ompletion (TGVAC).

Inspired by recent advancements in diffusion models for video generation (Ho et al., 2022a; Guo et al., 2024; Blattmann et al., 2023), we utilize a diffusion model to establish our baseline. Our approach follows a common two-phase training strategy: first, training a denoising UNet at the frame level to capture spatial features, and then incorporating motion training to ensure temporal coherence. Figure 3 illustrates the overall training scheme of our method.

**Frame-level Training**. Inspired by (Ozguroglu et al., 2024), we fine-tune a conditional diffusion model (e.g., Stable Diffusion (Rombach et al., 2022)) to perform frame-level amodal completion. Specifically, we optimize the following latent diffusion objective:

$$\min_{\theta_k} \mathbb{E} \left|\left| \epsilon - \epsilon_\theta(\mathbf{z}_t \oplus \mathcal{E}(\mathbf{m}_i), \mathcal{C}(y), t) \right|\right|_2^2 \tag{2}$$

Here, $\epsilon \sim \mathcal{N}(0, 1)$ represents Gaussian noise. The variable $\mathbf{z}_t$ denotes the noisy embedding of the amodal target object $\mathbf{o}$ at a given video frame. It is obtained by first encoding the video frame $\mathbf{x}_i$ as $\mathbf{z} = \mathcal{E}(\mathbf{x}_i)$ using the VAE encoder $\mathcal{E}$, and then adding embedding noise at time step $t \in [0, T]$, resulting in $\mathbf{z}_t = \mathcal{N}(\mathbf{z}, \mathcal{E}(\epsilon), t)$. The term $\mathcal{C}(y)$ represents the CLIP text embedding of the input prompt $y$. The function $\epsilon_\theta$ is a denoising U-Net, where $\theta_k$ represents the optimal frame-level parameters. The denoiser takes as input $\mathcal{E}(\mathbf{z}_t)$, $\mathcal{E}(\mathbf{m}_i)$, $\mathcal{C}(y)$, and $t$.

Specifically, let $L$ be the number of layers in $\epsilon_\theta$. Each layer feature is computed as follow:

$$\begin{aligned} f_0 &= Conv^{(0)}(\mathbf{z}_t + \mathcal{E}(\mathbf{m}_i)) \\ f_l &= \text{CrossAttn}(f_l, C(y)) \\ f_{l+1} &= Conv^{(l+1)}(f^l) \end{aligned} \tag{3}$$

At the first layer, $\mathbf{z}_t$ and $\mathcal{E}(\mathbf{m}_i)$ are concatenated and passed to a convolutional layers to enforce adherence to the visible parts of the object, ensuring the localization of the object need to be completed. Then, at each layer $l \in [0, L]$ of $\epsilon_\theta$, $\mathcal{C}(y)$ is incorporated through cross-attention, guiding the model towards reconstructing the complete object according to the text prompt $y$. Convolution layers and cross attention are designed similar as in ablated U-Net foloww (Rombach et al., 2022).

**Motion Training.** After pretraining at the frame level, we focus on modeling the motion dynamics between frames. This step aims to create smooth motion generation and enhance completion quality, allowing frames

with less occlusion to share features with those that have more occlusion. To leverage the knowledge from frame-level training, it is advantageous to inflate the network so that image layers can handle video frames independently (Guo et al., 2024).

Following recent approaches (Guo et al., 2024; Ho et al., 2022a; Blattmann et al., 2023), we modify the model to accept video tensors $\mathbf{v} \in \mathbb{R}^{B \times N \times C \times H \times W}$, where $B$ represents the batch axis and $N$ the time axis. Internally, when feature maps pass through image layers, the temporal axis $N$ is reshaped into the batch axis $B$, allowing the network to process each frame independently. After processing, the feature map is reshaped back into a 5D tensor before passing through motion modeling.

For motion modeling, we introduce module that processes temporal dynamics by reshaping the spatial axes $h$ and $w$ into the batch axis and reversing them after processing. Inspired by (Guo et al., 2024), we design this module using a Transformer architecture (Vaswani et al., 2017). After a frame-level layer, feature maps from video frames $f_l{}^{(i)}{}_{i=0}^{N} \in \mathbb{R}^{(b \times h' \times w') \times c'}$, where the spatial dimensions are merged into the batch axis, are obtained. We employ relative position embeddings to maintain the order of frames, enabling the temporal attention block to capture temporal coherence.

During motion training, we freeze the frame-level layers and train the motion module by optimizing the following objective:

$$\min_{\theta_m} \mathbb{E} \big|\big| \epsilon - \epsilon_\theta(\mathbf{z}'_t \oplus \mathcal{E}(\mathbf{v}_m), \mathcal{C}(y), t) \big|\big|_2^2 \tag{4}$$

Here, $\theta_m$ represents the parameters of the motion module. The variable $\mathbf{z}'_t$ denotes the noisy embedding of the amodal target object $\mathbf{o}$ at a given the entire video. It is obtained by first encoding the video $\mathbf{v}$ as $\mathbf{z}' = \mathcal{E}(\mathbf{v})$ using the VAE encoder $\mathcal{E}$, and then adding embedding noise at time step $t \in [0, T]$, resulting in $\mathbf{z}'_t = \mathcal{N}(\mathbf{z}, \mathcal{E}(\epsilon), t)$.

### 3.4 Zero-shot Inference for Long Video

While the above pipeline can theoretically handle videos of arbitrary length $N$, the model may experience significant quality degradation when generating videos longer than those used in training (Guo et al., 2024; Zhang et al., 2024). Inspired by approaches such as MultiDiffusion (Bar-Tal et al., 2023), which generates high-resolution images seamlessly composed of multiple patches, and Temporal Diffusion (Zhang et al., 2024), which handles video inpainting by generating multiple clips, we adapt Temporal Diffusion (Zhang et al., 2024) for our video amodal completion task. In specific, we divide the video into smaller training-sized clips $N'$, denoted as $\mathbf{v}^i$ for $i \in [1, N']$, using a stride $s$. At each denoising timestep $t$, our model is applied $N'$ times to produce $N'$ output clips, represented as

$$\mathbf{v}_{out_t}^i = \epsilon_\theta(\mathbf{z}_{t-1}, \mathcal{E}(\mathbf{v}^i), t-1, \mathcal{E}(\mathbf{v}_m^i), \mathcal{C}(y)) \tag{5}$$

Overlapping frames between these clips are averaged based on the number of times they are processed, ensuring consistency across the entire video sequence.

## 4 Experimental Result

### 4.1 Implementation Details

**Training.** Our model is built upon Stable Diffusion (Rombach et al., 2022) version 1.5 from the Diffusers library (von Platen et al., 2022). Initially, we train the frame-level layers of our U-Net using the Pix2Gestalt dataset (Ozguroglu et al., 2024), which consists of approximately 800,000 data samples. This stage involves 500,000 training steps with a batch size of 16, where the input image resolution is $256 \times 256$. We employ DDIM (Song et al., 2021) sampling with denoising steps $T = 1000$. Following frame-level training, we freeze the frame-level layers and proceed to train the motion layers using our synthesized dataset described in Section 3.2. The dataset is split 80-20 for training and validation. We conduct 500,000 training steps with a batch size of 16 and train on sequences of 14 frames per sample, each with a resolution of $256 \times 256$. Both

Table 2: **Quantitative results.** We compare our method against frame-level amodal completion methods Pix2gestalt Ozguroglu et al. (2024), ProgressiveAmodal Xu et al. (2024), and evaluate generated results using different metrics, including CLIP Radford et al. (2021) (high-level), LPIPS Zhang et al. (2018) (low-level), and Temporal Consistency Esser et al. (2023).

| Method | Easy Cases | | | | Hard Cases | | | |
|---|---|---|---|---|---|---|---|---|
| | CLIP↑ | LPIPS↓ | TC↑ | User Preference | CLIP↑ | LPIPS↓ | TC↑ | User Preference |
| ProgressiveAmodal Xu et al. (2024) | 0.88 | 0.14 | 0.92 | 0.10 | 0.86 | 0.18 | 0.85 | 0.11 |
| Pix2Gestalt Ozguroglu et al. (2024) | 0.89 | 0.12 | 0.92 | 0.12 | 0.86 | 0.17 | 0.87 | 0.08 |
| **TGVAC (Ours)** | **0.93** | **0.06** | **0.96** | **0.78** | **0.92** | **0.10** | **0.93** | **0.82** |

training stages use a learning rate of 0.0005. Our experiments are performed on eight NVIDIA RTX A6000 GPUs, each with 48 GB of memory.

**Inference.** During inference, we conduct zero-shot testing on arbitrary video sequences using Temporal Diffusion (Section 3.4) with a stride of $o = 4$ and clip length of 14 frames. We utilize DDIM (Song et al., 2021) with 30 denoising steps and apply a classifier guidance scale of 7.5.

## 4.2 Comparison with related methods

**Prior works.** To the best of our knowledge, no existing work has explored amodal completion at the video level. In this study, we benchmark TGVAC against frame-level amodal completion techniques to highlight its ability to maintain temporal consistency while ensuring high completion quality. We compare it with recent state-of-the-art (SOTA) frame-level amodal completion methods, specifically Pix2gestalt (Ozguroglu et al., 2024) and ProgressiveAmodal (Xu et al., 2024).

**Validation Set.** Our proposed synthetic dataset (Section 3.2) is divided into an 80-20 split for training and validation. The validation set comprises 1,400 videos for quantitative evaluation. Within this set, we classify video frames into easy cases (less than 50% occlusion) and hard cases (greater than 50% occlusion), following the approach outlined in (Xu et al., 2024).

**Metrics.** Following (Xu et al., 2024), we employ CLIP (Radford et al., 2021) for assessing high-level image similarity and LPIPS (Zhang et al., 2018) for evaluating low-level image similarity. Additionally, we measure Temporal Consistency (TC) by calculating the cosine similarity between consecutive frames within the CLIP-Image feature space, as done in (Esser et al., 2023; Zhang et al., 2024). To further measure perceptual video quality, we conduct a user preference study involving 20 university students. Participants are shown 20 input videos (10 easy cases, 10 hard cases) alongside generated amodal completion videos from each method, and they are asked to vote for the video that appears most complete and realistic. Detailed study protocols and user demographics are provided in B.

**Quantitative comparison.** Table 2 demonstrate that in both 'Easy' and 'Hard' scenarios (described in Section 4.2), TGVAC consistently surpasses Pix2Gestalt and ProgressiveAmodal. When evaluated using the high-level image similarity metric CLIP, our approach outperforms both baselines, achieving scores of 0.93 compared to 0.89 in easy cases and 0.92 compared to 0.86 in hard cases. Although these improvements appear modest, this is expected in the task of amodal completion because even suboptimal generated parts often align well with the visible sections' colors and shapes, resulting in high CLIP similarity scores. In contrast, significant advancements are evident in the low-level image similarity metric LPIPS, where TGVAC shows clear superiority: 0.06 versus 0.12 in the 'Easy' case and 0.10 versus 0.17 in the 'Hard' case. This improvement is attributed to the training with the motion module, which enhances coherence between frames and minimizes abrupt changes during occlusion. This advantage is further highlighted by the Temporal Consistency metric, where TGVAC also outperforms the other two, scoring 0.96 versus 0.92 in the 'Easy' case and 0.93 versus 0.87 in the 'Hard' case. In the User Preference study, TGVAC achieves significantly higher preference scores in both "Easy Cases" (0.78) and "Hard Cases" (0.82), substantially outperforming both Pix2Gestalt (0.10 and 0.11) and ProgressiveAmodal (0.12 and 0.08). These results quantitatively

demonstrate that users overwhelmingly preferred our approach for its superior completeness, realism, and perceptual quality in both straightforward and challenging scenarios.

**Qualitative comparison.** Figure 4 and Figure 5 jointly illustrate the qualitative advantage of TGVAC over frame-level baselines under two occlusion regimes. Each figure shows five evenly-spaced frames (columns) of a video, while the rows list: the source frames, the visible frames, the amodal completion ground truth (GT), ProgressiveAmodal Xu et al. (2024) (PA) results, Pix2gestalt Ozguroglu et al. (2024) (Pix2ges) results, our ablated variant without motion training, and the full TGVAC. Regarding the easy case, both baselines recover only fragments of the car and exhibit broken contours or missing wheels. Meanwhile, the motion-free variant although predict plausible completion at each frame, these predictions are not consistent throughout the video. For instance, the completed car in the last frame looks quite different compare to the car in other frames due to large occlusion. By exploiting temporal cues, TGVAC reconstructs the entire vehicle that closely matches the GT and remains temporally coherent across all frames. Regarding the hard case, here more than half of a hiking person is obscured by terrain in several frames. PA and Pix2gestalt fail to predict the missing torso and limbs, producing implausible thin shapes. Meanwhile, our no-motion variant suffers from inconsistency and faint boundaries. In contrast, TGVAC still infers a complete, well-shaped human figure (including backpack and limbs) and preserves appearance consistency as the person moves. Together, these examples highlight that integrating motion information enables TGVAC to deliver more complete and temporally stable amodal completions, with especially pronounced gains when occlusion is severe.

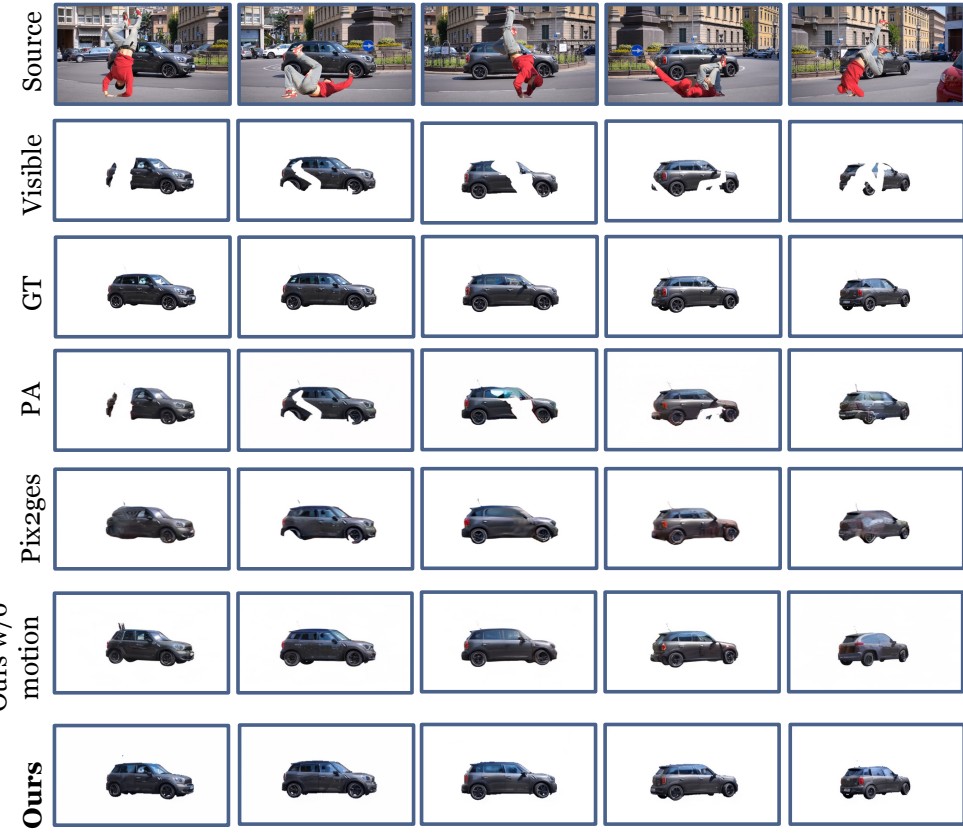

Figure 4: **Qualitative comparison on Easy Case (less than 50% occlusion).** We compare TGVAC against frame-level amodal completion methods Pix2gestalt (Ozguroglu et al., 2024), ProgressiveAmodal (PA) (Xu et al., 2024).

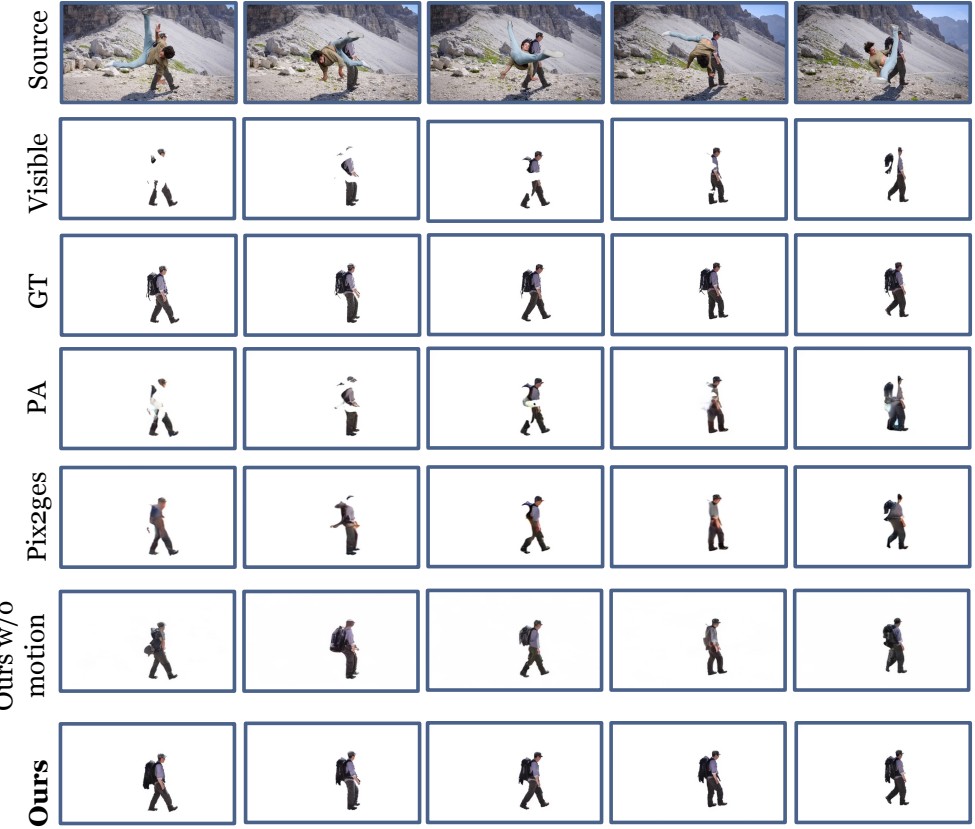

Figure 5: **Qualitative Comparison on Hard case (greater than 50% occlusion.)** We compare TGVAC against frame-level amodal completion methods Pix2gestalt (Ozguroglu et al., 2024), ProgressiveAmodal (PA) (Xu et al., 2024), and TGVAC w/o motion training.

### 4.3 Zero-shot Inference on Natural Videos.

In Figure 6, we present qualitative results demonstrating zero-shot performance on natural videos. As shown, our model effectively completes amodal occlusions in real-world video sequences. This highlights the value of our synthetic amodal completion dataset, which captures diverse object types, textures, motions, and scenarios, enabling robust zero-shot transfer to natural video contexts. In particular, as shown in Figure 6, in the top subfigure, TGVAC successfully reconstructs the occluded parts of the monkey, producing a temporally consistent and realistic view of the animal's full shape throughout its motion. Similarly, in the bottom left subfigure, TGVAC exhibits perceptual realism by completing the fish's occluded body parts with smooth motion across frames, maintaining visual consistency.

### 4.4 Ablation Study

**Effect of text prompt, motion training, and temporal diffusion** Table 3 presents an ablation study that examines three components of TGVAC: (1) whether a text prompt is used, (2) whether motion training is applied, and (3) whether temporal diffusion is included. Note that temporal diffusion can only be applied when motion training is used.

By examining each row of the table, we see how enabling or disabling the text prompt, motion training, and temporal diffusion affects the performance in both easy and hard scenarios. Including a text prompt typically

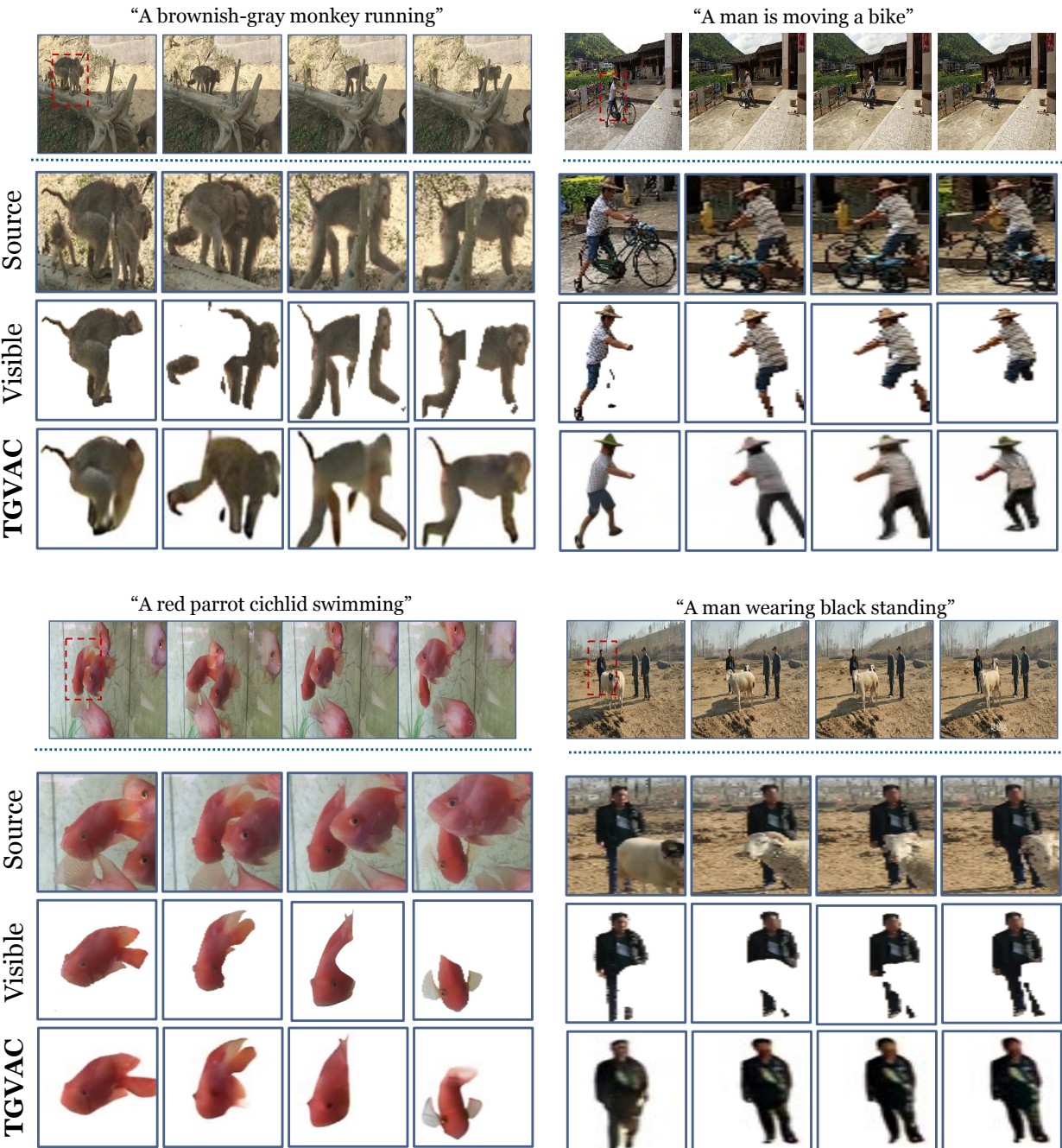

Figure 6: Qualitative results showing zero-shot amodal completion on natural videos.

improves CLIP, as it provides strong semantic guidance. Motion training often leads to lower LPIPS, indicating higher perceptual fidelity, particularly in challenging motion sequences. Temporal diffusion addresses long-video consistency, often improving TC by reducing flicker or abrupt changes. In easier cases, partial combinations of these techniques can still yield respectable results, but in harder cases, where more complex movements or scene details are involved, the differences among the ablations become more pronounced. In general, configurations that include all three components—text prompt, motion training, and temporal

Table 3: Ablation Study for Text Prompt, Motion Training, and Temporal Diffusion

| Text prompt | Motion Training | Temporal Diffusion | Easy Cases | | | Hard Cases | | |
|---|---|---|---|---|---|---|---|---|
| | | | CLIP↑ | LPIPS↓ | TC↑ | CLIP↑ | LPIPS↓ | TC↑ |
| ✗ | ✗ | ✗ | 0.88 | 0.12 | 0.91 | 0.86 | 0.17 | 0.88 |
| ✓ | ✗ | ✗ | 0.89 | 0.11 | 0.92 | 0.87 | 0.15 | 0.88 |
| ✗ | ✓ | ✗ | 0.91 | 0.08 | 0.94 | 0.91 | 0.12 | 0.91 |
| ✓ | ✓ | ✗ | 0.92 | 0.08 | 0.95 | 0.91 | 0.12 | 0.92 |
| ✗ | ✓ | ✓ | 0.92 | 0.07 | 0.96 | 0.91 | 0.11 | 0.93 |
| ✓ | ✓ | ✓ | 0.93 | 0.06 | 0.96 | 0.92 | 0.10 | 0.93 |

Table 4: **Effect of temporal diffusion stride for long video inference.** Smaller strides ($o = 2$ or $o = 4$) achieve the best high-level similarity, low-level detail, and temporal consistency, while larger strides lead to reduced performance.

| TD Stride | CLIP↑ | LPIPS↓ | TC↑ |
|---|---|---|---|
| $o = 2$ | 0.92 | 0.09 | 0.94 |
| $o = 4$ | 0.92 | 0.08 | 0.94 |
| $o = 8$ | 0.91 | 0.10 | 0.92 |

diffusion—have a tendency to achieve better semantic alignment (higher CLIP), better perceptual quality (lower LPIPS), and improved temporal coherence (better TC), highlighting the contribution of TGVAC.

**Effect of Temporal Diffusion for Long Video Inference** Table 4 explores the effect of different temporal diffusion strides (TD stride) on long video inference. The TD stride represents the overlap between video frames during inference, with $o = N$ indicating no overlap (i.e., no use of temporal diffusion). For strides $o = 2$ and $o = 4$, the performance is the highest, with both achieving similar CLIP score (0.92), low LSIPS (0.09 and 0.08), and strong Temporal Consistency (TC) at 0.94. This suggests that these strides provide a good balance between maintaining high-level image similarity, low-level detail, and temporal consistency. When the stride increases to $o = 8$, the performance drops slightly, with a CLIP score of 0.91 and a slight increase in LSIPS to 0.10, indicating a decline in high-level similarity and low-level detail. TC also drops to 0.92, showing reduced temporal stability. These results demonstrate that smaller strides ($o = 2$ or $o = 4$) yield the best temporal consistency and image similarity, while larger strides or no overlap result in a decline in video quality and temporal stability. We opt for $o = 4$ because it requires less computation than $o = 2$.

### 4.5 Comparison with video amodal completion methods

We further investigate the efficacy of our model for video amodal segmentation task. We perform zero-shot evaluation of TGVAC on FISHBOWL Tangemann et al. (2021) and MOViD-A Gao et al. (2023) datasets. We follow previous work settings Gao et al. (2023); Yao et al. (2022) to use $mIoU$ and $mIoU_{occ}$ (occluded mIoU) as evaluation metrics. In fact, given a visible-amodal sequence pair in each frame, where the ground-truth visible mask is $\mathcal{M}_i$, and the predicted and ground-truth amodal masks are $\hat{\mathcal{A}}_i$ and $\mathcal{A}_i$, respectively, mIoU is defined as $\frac{\hat{\mathcal{A}}_i \cap \mathcal{A}_i}{\hat{\mathcal{A}}_i \cup \mathcal{A}_i}$ and mIoU$_{occ}$ as $\frac{(\hat{\mathcal{A}}_i - \mathcal{M}_i) \cap (\mathcal{A}_i - \mathcal{M}_i)}{(\hat{\mathcal{A}}_i - \mathcal{M}_i) \cup (\mathcal{A}_i - \mathcal{M}_i)}$. We report the mean values across all frames in the dataset as mIoU and mIoU$_{occ}$. Table 5 shows the meanIoU metrics of TGVAC and other baselines (i.e. C2F-Seg Gao et al. (2023) and SaVos Yao et al. (2022)) on FISHBOWL Tangemann et al. (2021) and MOViD-A Gao et al. (2023) datasets. Our method shows competitive results with SOTA methods, which are directly supervised on these two datasets.

Table 5: Quantitative results on **FISHBOWL** and **MOViD-A**. We report and compare Mean-IoU metrics of C2F-Seg with baselines.

| Methods | Train Setting | FISHBOWL | | MOViD-A | |
|---|---|---|---|---|---|
| | | $mIoU_{\text{full}}$ | $mIoU_{\text{occ}}$ | $mIoU_{\text{full}}$ | $mIoU_{\text{occ}}$ |
| SaVos | Self-supervised | 88.63 | 71.55 | 60.61 | 22.64 |
| C2F-Seg | Supervised | 91.68 | 81.21 | 71.67 | 36.13 |
| **TGVAC (Ours)** | Zeroshot | **90.34** | **78.22** | **68.12** | **32.74** |

### 4.6 Comparison with text-to-video diffusion pretrained

We conduct experiments adopting pretrained video diffusion models for the task of video amodal completion. We leverage state-of-the-art CogVideoX Yang et al. (2024). Since CogVideoX is originally designed to generate videos from text alone, we modified it to handle video amodal completion by incorporating additional inputs: the input video and its corresponding visible-region mask. We encoded the video and mask features and concatenated them with the noise vector used during diffusion, allowing the model to condition on both spatial and temporal context. We refer to this modified version as CogVideoX-Amodal. Table 6 shows the comparison between our TGVAC and CogVideoX-Amodal. While pretrained video diffusion models like CogVideoX are powerful, they are not optimized for amodal video completion tasks like ours. As shown in the aforementioned table, our TGVAC achieves better performance across metrics. This is due to the nature of the amodal completion task: unlike general video synthesis from text, amodal completion requires precise preservation of visible content and plausible generation of occluded regions. Our TGVAC is trained with these constraints in mind, leveraging image-based amodal completion knowledge and motion modeling to maintain visual fidelity and temporal coherence.

Table 6: Comparison with text-to-video diffusion pretrained.

| Method | CLIP↑ | LPIPS ↓ | TC ↑ |
|---|---|---|---|
| CogVideoX-Amodal | 0.87 | 0.16 | 0.89 |
| TGVAC (Ours) | 0.92 | 0.08 | 0.94 |

## 5 Conclusion & Discussion

**Conclusion.** In this work, we have explored the challenging problem of video amodal completion. To this end, we have introduced a synthetic dataset, equipped with detailed descriptions of objects of interest, enabling effective zero-shot transfer to natural video scenarios. Additionally, we proposed a diffusion-based, text-guided video amodal completion framework enhanced with a motion continuity module to ensure temporal consistency across video frames. By incorporating temporal diffusion, we managed long video sequences with improved inference accuracy and coherence. As the first research in video amodal completion, this work plays the role of a baseline for advancing video amodal completion and its applications in video editing, analysis, and beyond.

**Discussion.** Video amodal completion is an emerging field that extends the concept of image amodal completion into the video domain. While the current results mark an important milestone, the field remains in its infancy, with substantial opportunities for further exploration and refinement. Future research should focus on expanding the scope of video amodal completion to tackle more complex and challenging real-world application such as video object tracking in occlusion environments, 3D video reconstruction, virtual and augmented reality.

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

# A    Additional Qualitative Results

## A.1    Additional Zero-shot Inference on Natural Videos

Figure 7 present additional qualitative results demonstrating zero-shot performance on natural videos with real occlusion. These results highlight the robustness and adaptability of our method, producing realistic and temporally coherent amodal completions in natural, unseen video contexts.

## A.2    Additional Qualitative Comparison

In Section 4.2, we compare our work with prior related works, specifically Pix2gestalt Ozguroglu et al. (2024) and ProgressiveAmodal Xu et al. (2024). This section provides additional qualitative comparisons between our work and these related works, as shown in Figure 8 to 10. Overall, our method showcases its advantage in delivering realistic, perceptually complete reconstructions with superior temporal alignment, which are critical for video amodal completion tasks. Notably, the reconstructed details exhibit higher realism and fidelity, maintaining the object's structure. Our approach also demonstrates remarkable robustness in challenging scenarios where objects are heavily occluded or undergo significant temporal transformations. Compared to the baselines, our method produces outputs with enhanced temporal consistency and smoother motion continuity.

For example, Figure 9 showcase an owl standing, the ground truth once again provides a reference for ideal performance. The proposed method surpasses alternatives by preserving the goat's shape and completing occluded regions with higher perceptual realism, maintaining a consistent temporal flow across frames. In contrast, PA and Pix2gestalt introduce distortions, such as inconsistent shapes or unnatural blending of occluded areas. Similarly, in Figure 10 while competing methods struggle with temporal flickering and incomplete reconstructions of occluded details, the proposed method excels in maintaining realistic outlines, smooth transitions, and fine details like the person's arm.

# B    Additional Experiment Details

## B.1    User study

To measure perceptual video quality more effectively, we conducted a user preference study using a custom-designed interface. The study involved 20 university students (8 graduate students, 12 undergraduate students) from different ethnicities and academic background. Participants were tasked with evaluating the results of different amodal completion methods, focusing on how complete and realistic the generated outputs appeared.

As shown in Figure 11, the interface displayed five videos side by side: the input video, ground truth, and outputs from three different methods: Pix2gestalt Ozguroglu et al. (2024), ProgressiveAmodal Xu et al. (2024), and our proposed approach. For each sample, the results from these methods were randomly shuffled and displayed as Method 1, Method 2, and Method 3. Each participant was instructed to review the videos and make a selection based on which generated result they found most complete and realistic. The ground truth video was provided as a reference for participants to understand the ideal completion outcome. Participants were shown a set of 20 video, evenly divided between 10 easy cases (0-50% occlusion) and 10 hard cases ( 50% occlusion). For each set, participants viewed the input video to understand the scene and degree of occlusion. They then observed the results from each method alongside the ground truth. Using the interface, they selected the video they considered most satisfactory in terms of perceptual completeness and realism. Participants were guided to focus on: (i) perceptual completeness: How well the occluded parts were reconstructed, ensuring consistency in appearance; (ii) realism: The natural flow and appearance of motion throughout the video; (iii) temporal consistency: The smooth transition across video frames without artifacts or noticeable disruptions. The study was conducted in a controlled environment with consistent lighting and display settings. After each selection, participants confirmed their choices by clicking the Select button below the chosen method. Results were logged automatically, ensuring randomization of video order to reduce selection bias.

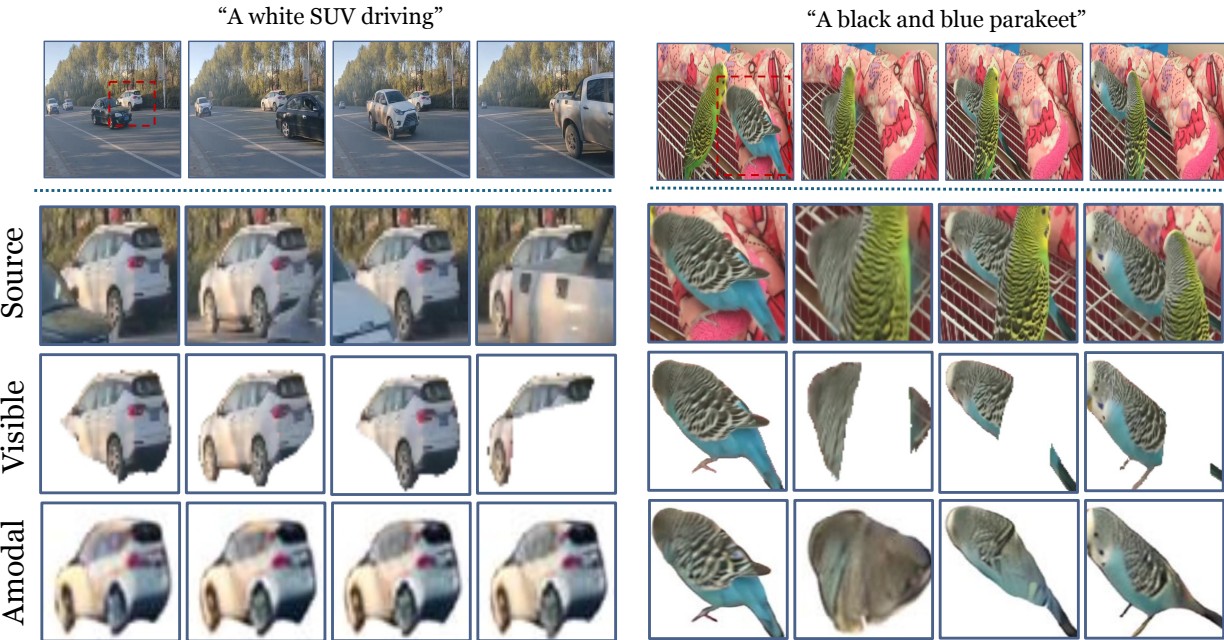

Figure 7: **Additional Qualitative Results 3** showing zero-shot amodal completion on natural videos. Best viewed in zoom and color.

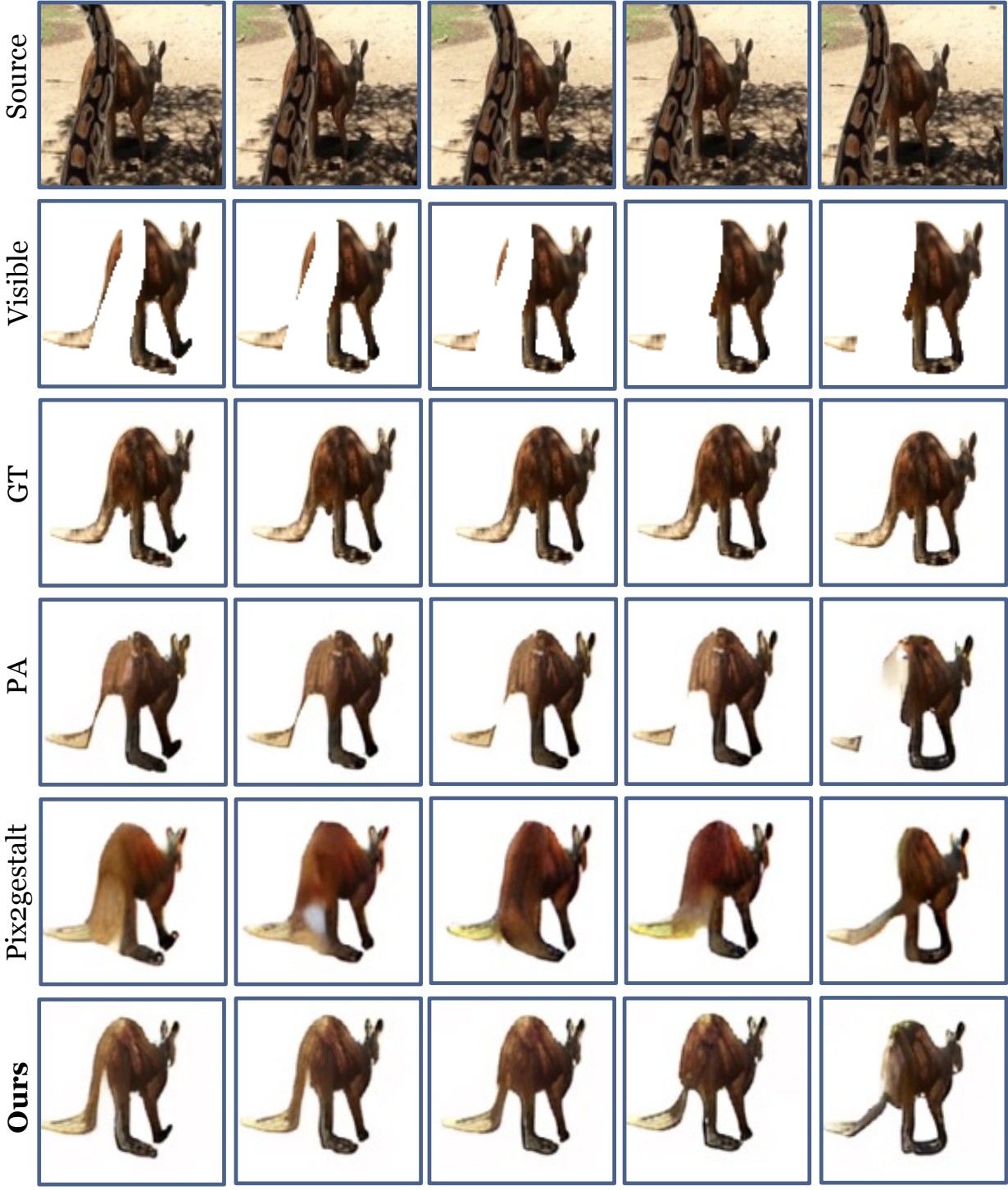

Figure 8: **Additional Qualitative Comparison 1.**

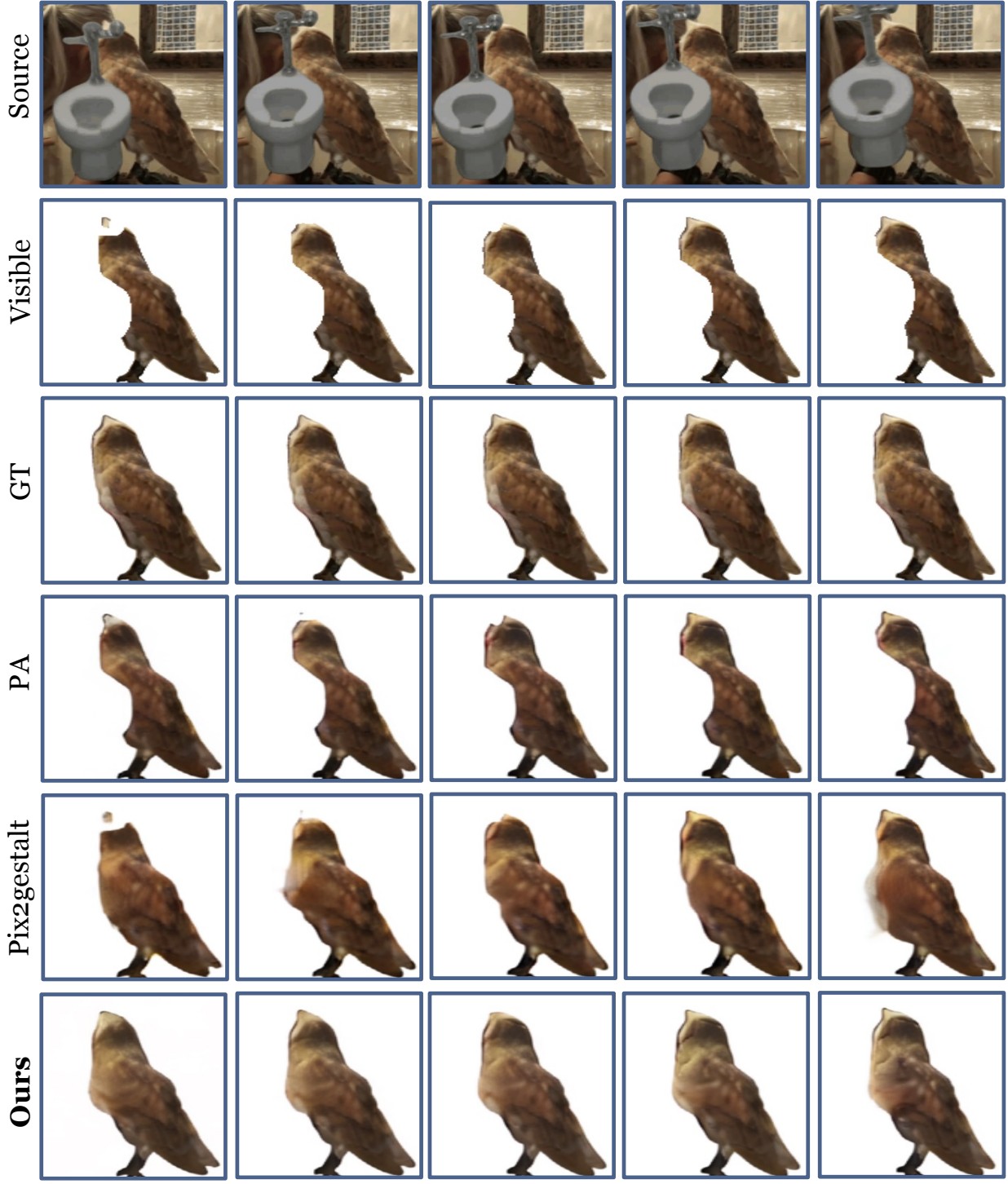

Figure 9: **Additional Qualitative Comparison 2.**

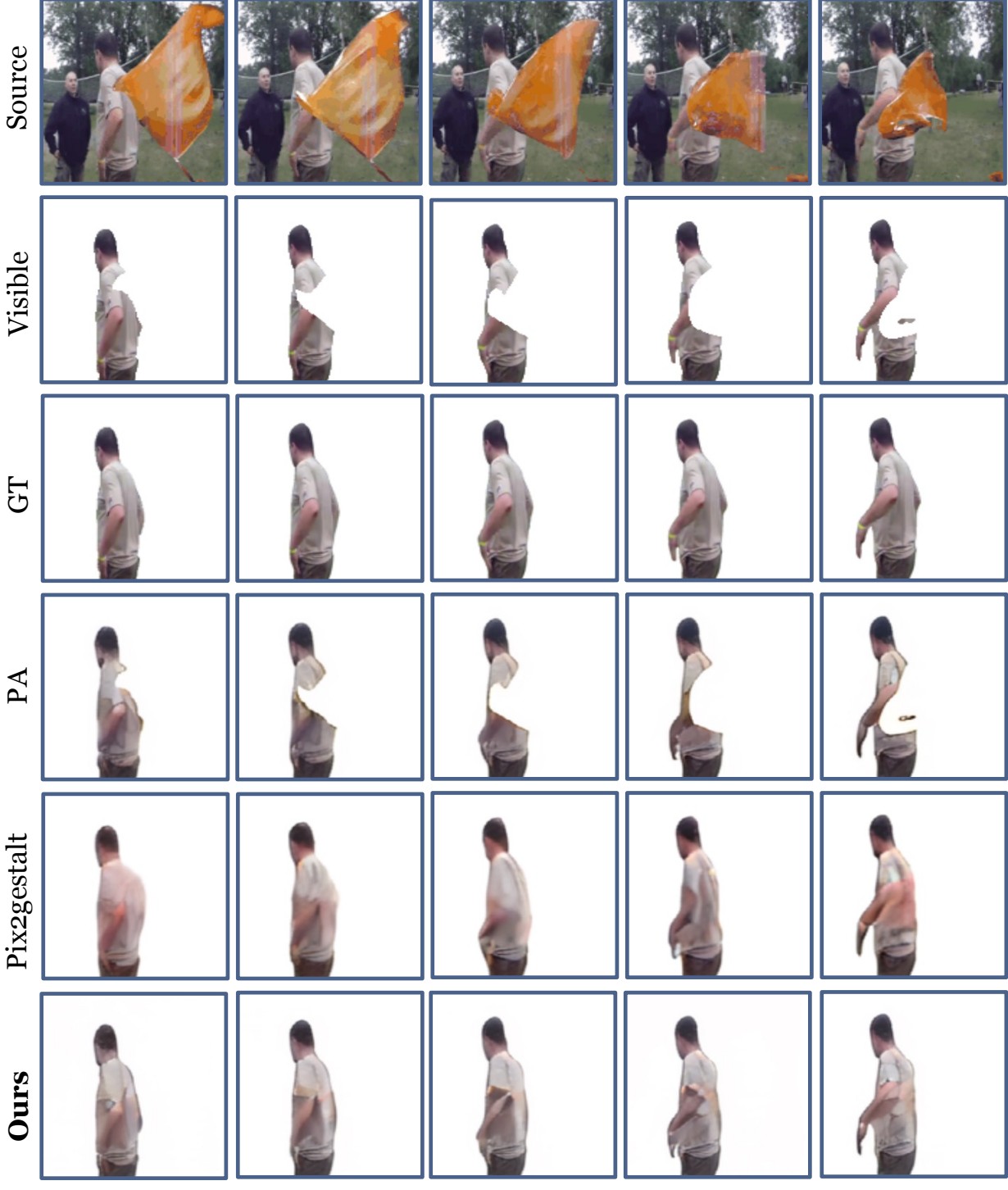

Figure 10: **Additional Qualitative Comparison 3.**

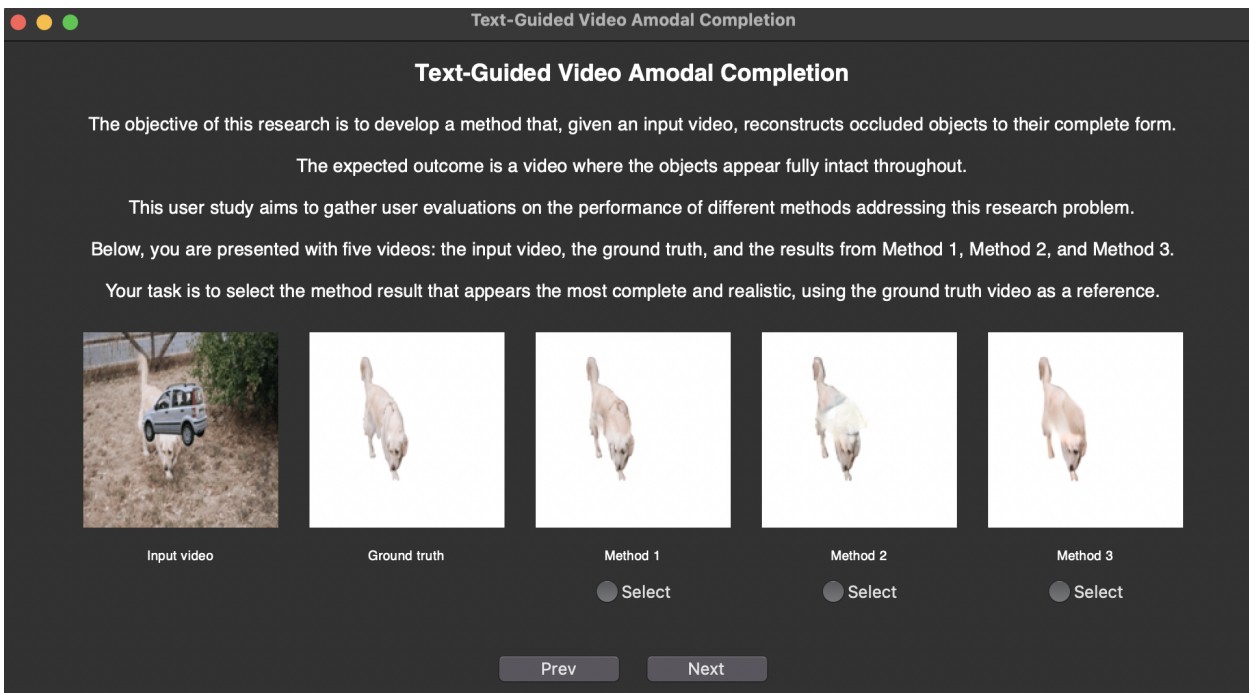

Figure 11: User interface used for the user preference study. Participants were asked to select the method result that appeared most complete and realistic. The three methods evaluated were Pix2gestalt Ozguroglu et al. (2024), ProgressiveAmodal Xu et al. (2024), and our proposed approach. For each sample, the results from these methods were randomly shuffled and displayed as Method 1, Method 2, and Method 3.

