# OpenReview forum: "Text-Guided Video Amodal Completion"
_TMLR — Rejected by TMLR_

### Review · Reviewer_JU1P · 2025-05-05

**Summary Of Contributions:**

This paper introduces a framework for text-guided video amodal completion, which aims to recreate complete objects from partially obscured views in videos. The authors make three main contributions: (1) a synthetic dataset for video amodal completion with textual descriptions to guide the completion process, (2) a diffusion-based model that uses a two-phase training strategy to generate complete object shapes, textures, and motions, and (3) zeroshot evaluation on longer videos by dividing them into smaller clips and then integrating the results.

**Audience:**

Yes

**Claims And Evidence:**

Yes

**Requested Changes:**

Please consider the weaknesses. I believe point one is particularly important since I am not sure right now what components are mainly contributing to the performance. For point 2, I am a bit concerned regarding what I feel is low temporal variation. But I will prefer to see what the other reviewers think on this. But I would like to know from the authors on why long video datasets like LVOS were not considered.

**Strengths And Weaknesses:**

Strengths:

1. The paper is well-written and easy to understand.
2. The problem is well-motivated. There is a lack of datasets for the video completion task, and the paper addresses this gap.
3. The qualitative and quantitative results show that the proposed model outperforms baselines on the given tasks.


Weaknesses:

1. The ablations in table 3 is confusing. Its not really clear on whats the impact of each component individually. Could the authors provide a different ablation table where each component is removed once with the other two present, and final row of all components present? More specifically a total of four rows in the ablation table: 1) Text prompt + Motion Training; 2) Text prompt + Temporal Diffusion; 3) Motion Training + Temporal Diffusion; 4) Text prompt + Motion Training + Temporal Diffusion. 3 and 4 are already present in Table 3. Could the authors provide 1 and 2 as well? Because right now the improvements seem to be a bit incremental and its not really clear what components are contributing to those improvements.

2. From the given qualitative examples, I feel that there is not much temporal variations in the produced synthetic data. Why did the authors not consider datasets such as LVOS [1], which are long videos for video object segmentation task. This would be a much better base dataset to produce the synthetic data from.

[1] LVOS: A Benchmark for Long-term Video Object Segmentation

---

> ### Author Response · Authors · 2025-05-21
> **Response to reviewer JU1P**
>
> We are grateful to the reviewer for your constructive feedback. We would like to address the reviewer’s concerns as follow:
>
> ## 1. The ablations in table 3 is confusing. Its not really clear on whats the impact of each component individually. Could the authors provide a different ablation table where each component is removed once with the other two present, and final row of all components present? More specifically a total of four rows in the ablation table: 1) Text prompt + Motion Training; 2) Text prompt + Temporal Diffusion; 3) Motion Training + Temporal Diffusion; 4) Text prompt + Motion Training + Temporal Diffusion. 3 and 4 are already present in Table 3. Could the authors provide 1 and 2 as well? Because right now the improvements seem to be a bit incremental and its not really clear what components are contributing to those improvements.
>
> Response: We thank the reviewer for the suggestion with the two suggested additional experiments: (1) (text prompt + motion training) and (2) (text prompt + temporal Diffusion). For experiment (1), we conduct the experiment as in the following table. We also have added in the revised manuscript the experiment (1) in our ablation study table 3 (Page 12).
> | Text prompt | Motion Training | Temporal Diffusion | CLIP↑ (Easy) | LPIPS↓ (Easy) | TC↑ (Easy) | CLIP↑ (Hard) | LPIPS↓ (Hard) | TC↑ (Hard) |
> |-------------|-----------------|--------------------|--------------|---------------|------------|--------------|---------------|------------|
> | ✓           | ✓               | ✗                  | 0.92         | 0.08          | 0.95       | 0.91         | 0.12          | 0.92       |
>
>
> For experiment (2), this configuration is not feasible within our current framework. Temporal diffusion is designed to operate on top of motion training. Therefore, we cannot evaluate the temporal diffusion component in isolation without motion training.
>
>
> ## 2. From the given qualitative examples, I feel that there is not much temporal variations in the produced synthetic data.
>
>
> Response: We thank the reviewer's concerns. In response, we have revised Figure 4 and Figure 5 in the manuscript to include clips where the target objects exhibit more dynamic motion.
>
> Moreover, we introduce the following table to better illustrate the temporal complexity of occlusions in our dataset. Particularly, we compute an occlusion-coverage ratio to quantify the spatial spread of an occluder over the entire video over time. For each frame, $i\in\{1,\dots,N\}$, the binary intersection $I_i = A_{i}\cap O_{i}$ between the main‑object amodal mask $A_{i}$ and the occluder mask $O_i$; we then form the per‑video unions $U_{\text{occ}} = \bigcup_{i=1}^{N} I_i$ and $U_{\text{obj}} = \bigcup_{i=1}^{N} A_{i}$.  Then the occlusion-coverage ratio for each video is computed as: $\frac{\lvert U_{\text{occ}} \rvert}{\lvert U_{\text{obj}} \rvert}$, where $\lvert\cdot\rvert$ denotes the cardinality (number of foreground pixels).  An occlusion-coverage ratio of $0$ indicates that the occluder never overlaps the object, while an occlusion-coverage ratio of $1$ means that every object pixel is hidden at least once during the sequence. This metric offers an indication of how much the occluder interacts with the object spatially and temporally throughout the sequence. This is added to the revised manuscript in section 3.2 page 6.
>
> |   | DAVIS | YTVIS | LASOT |
> | ------------- |:-------------:|:-------------:|:-------------:|
> | average occlusion-coverage rate      | 0.40 | 0.37| 0.42|
>
>
> ## 3.  Why did the authors not consider datasets such as LVOS [1], which are long videos for video object segmentation task. This would be a much better base dataset to produce the synthetic data from.
>
> Response: We thank the reviewer for the suggestion. One of our primary sources is LaSOT, which already consists of long‐term video sequences with significant object motion and variability It is important to note that LVOS was constructed by annotating a subset of LaSOT videos with pixel-level masks. We chose to build on LaSOT rather than LVOS because LaSOT offers broader object category coverage, which is important for improving generalization to diverse, real-world scenarios.

---

### Review · Reviewer_FHRB · 2025-05-05

**Summary Of Contributions:**

This paper introduces the task of text-guided video amodal completion, aiming to recover the full appearance of occluded objects in video using a visible mask and a textual prompt. The authors propose a two-stage framework combining a frame-level latent diffusion model with a motion-aware temporal module, trained on a synthetic dataset of 5,000 composited videos with text descriptions. The model outperforms image-based baselines and shows promising qualitative generalization to natural videos.

**Audience:**

Yes

**Broader Impact Concerns:**

No broader impact concerns.

**Claims And Evidence:**

Yes

**Requested Changes:**

1. Include more dynamic occlusion scenarios to better evaluate the model’s temporal reasoning. For example, consider cases where a moving object is occluded in different ways across frames, so that different parts become visible at different times. This setup would more clearly test whether the model can aggregate partial visual evidence over time. One illustrative (but NOT required) example could involve videos with dense text, such as a book or computer screen, where different portions of the text are occluded in each frame. This would highlight the model’s ability to fuse information temporally and recover structured content.
2. Report implementation details (resolution, object scale, model initialization) to improve reproducibility.
3. Include the frame-level-only model in qualitative results for fair comparison with image-based baselines.

**Strengths And Weaknesses:**

### strength
1. Novel and well-scoped task: The paper defines a new task—text-guided video amodal completion—which meaningfully extends amodal reasoning into video and vision-language settings, with relevance for video editing and embodied perception.

2. Clear architecture with strong results: The two-stage pipeline (frame-level diffusion + temporal module) is well structured and validated through ablations. The method outperforms image-based baselines (e.g., PA, Pix2Gestalt) on standard reconstruction metrics.

3. Systematic dataset construction and promising generalization: The synthetic dataset—generated via compositional occlusion and text prompting—offers a scalable training setup. Despite some limitations (to be shown in next section), the dataset enables the model to generalize qualitatively to natural videos.

### weakness
1. Temporal modeling is not meaningfully evaluated due to static object motion: Most training and evaluation examples involve static or slowly moving objects, limiting the opportunity to test temporal aggregation. For instance, in Figure 5, the object remains static and the occluded region remains hidden across all frames, making temporal modeling unnecessary. As a result, the motion module is underutilized, and performance gains from it (Table 2, Table 4) are modest.

2. Blurry outputs and missing implementation details: Completions often lack fine detail and texture. Additionally, the paper does not report important experiment settings such as  (1) frame resolution during training/inference, (2)  object size or occlusion ratio statistics (3) whether the diffusion backbone was pretrained or trained from scratch. These omissions limit reproducibility and hinder interpretation of visual quality.

3. Limited task generalization and lack of real-world evaluation: The method is tailored to a specific input setting (video + mask + text) and is not demonstrated on related tasks like image-level amodal segmentation or general video inpainting. Moreover, while qualitative results on natural videos are shown, there is no quantitative evaluation, which weakens the generalization claims.

4. Qualitative comparisons are not fully aligned: The paper only compares against image-based methods (PA, Pix2Gestalt). A more informative comparison would also include the frame-level-only variant of the proposed model (before motion training) to fairly assess the contribution of the temporal module. (e.g. in figure 4 & 5)

---

> ### Author Response · Authors · 2025-05-21
> **Response to Reviewer FHRB - Part 1**
>
> We appreciate the reviewer for your constructive feedback. We would like to address the reviewer’s concerns as follows:
>
> ## 1. Temporal modeling is not meaningfully evaluated due to static object motion: Most training and evaluation examples involve static or slowly moving objects, limiting the opportunity to test temporal aggregation. For instance, in Figure 5, the object remains static and the occluded region remains hidden across all frames, making temporal modeling unnecessary. As a result, the motion module is underutilized, and performance gains from it (Table 2, Table 4) are modest.
>
> Response: We thank the reviewer for the thoughtful comment. We agree that the original examples may not fully demonstrate the benefits of temporal modeling due to limited object motion.
>
> Your concern is addressed following in two terms:
>
> 1. Object Motion:
> To address this concern, we have revised Figure 4 and Figure 5 in the manuscript to include clips where the target objects exhibit more dynamic motion.
>
> Moreover, we introduce the following table to better illustrate the temporal complexity of occlusions in our dataset. Particularly, we compute an occlusion-coverage ratio to quantify the spatial spread of an occluder over the entire video over time. For each frame, $i\in\{1,\dots,N\}$, the binary intersection $I_i = A_{i}\cap O_{i}$ between the main‑object amodal mask $A_{i}$ and the occluder mask $O_i$; we then form the per‑video unions $U_{\text{occ}} = \bigcup_{i=1}^{N} I_i$ and $U_{\text{obj}} = \bigcup_{i=1}^{N} A_{i}$.  Then the occlusion-coverage ratio for each video is computed as: $\frac{\lvert U_{\text{occ}} \rvert}{\lvert U_{\text{obj}} \rvert}$, where $\lvert\cdot\rvert$ denotes the cardinality (number of foreground pixels).  An occlusion-coverage ratio of $0$ indicates that the occluder never overlaps the object, while an occlusion-coverage ratio of $1$ means that every object pixel is hidden at least once during the sequence. This metric offers an indication of how much the occluder interacts with the object spatially and temporally throughout the sequence. This is added to the revised manuscript in section 3.2 page 6.
>
> |   | DAVIS | YTVIS | LASOT |
> | ------------- |:-------------:|:-------------:|:-------------:|
> | average occlusion-coverage rate      | 0.40 | 0.37| 0.42|
>
>
> 2. Performance Gain:
> Regarding the modest quantitative gains: Although these improvements appear modest, this is expected in the task of amodal completion because even poor completion parts still can align with the visible sections’ colors and shapes, resulting in high CLIP similarity scores. In contrast, the benefits of temporal modeling are more clearly reflected in low-level image similarity metrics such as LPIPS, where TGVAC shows clear superiority: 0.06 versus 0.12 in the ‘Easy’ case and 0.10 versus 0.17 in the ‘Hard’ case. This improvement is attributed to the training with the motion module, which enhances coherence between frames and minimizes abrupt changes during occlusion. We have discussed this in our original submitted manuscript (Page 8, quantitative comparison subsection).

---

> ### Author Response · Authors · 2025-05-21
> **Response to Reviewer FHRB - Part 2**
>
> We appreciate the reviewer for your constructive feedback. We would like to address the reviewer’s concerns as follows:
> ## 2. Blurry outputs and missing implementation details: Completions often lack fine detail and texture. Additionally, the paper does not report important experiment settings such as (1) frame resolution during training/inference, (2) object size or occlusion ratio statistics (3) whether the diffusion backbone was pretrained or trained from scratch. These omissions limit reproducibility and hinder interpretation of visual quality.
>
> Response: We thank the reviewer for raising these concerns. We address the reviewer’s concerns in two parts: Blurry Output, and Implementation Details.
>
> 1. Blurry Output:
> The blurry outcomes is primarily due to computational constraints, which required us to resize all training and inference frames to 256 × 256. This inevitably softens fine details and textures, but the completions still meet our amodal‑completion goal: the synthesized, previously occluded regions blend naturally with the visible context and avoid physically implausible shapes. We believe that training at higher resolution on more powerful hardware could further enhance texture fidelity.
>
> 2. Implementation details:
> Frame resolution. Training and inference are both at 256 × 256 (We have mentioned this in implementation detail section 4.1, page 7).
> Object size / occlusion ratio. The following table shows the object size and occlusion ratio for each dataset source in our synthetic dataset. These are reported in the updated Table 1 (page 5) of the revised manuscript.
> |   | DAVIS | YTVIS | LASOT |
> | ------------- |:-------------:|:-------------:|:-------------:|
> | average object size      | 12.80 | 13.61 | 10.31|
> | average occlusion ratio      | 0.29 |  0.39 | 0.35|
> Pre‑training. Our U‑Netis initialized from Stable Diffusion v1.5 via via the HuggingFace Diffusers library. We then pre‑train the frame‑level layers on the Pix2Gestalt dataset (Ozguroglu et al., 2024), which provides roughly 800 k training samples. (We have mentioned this in implementation detail section 4.1, page 7).
>
> ### 3. Limited task generalization and lack of real-world evaluation: The method is tailored to a specific input setting (video + mask + text) and is not demonstrated on related tasks like image-level amodal segmentation or general video inpainting. Moreover, while qualitative results on natural videos are shown, there is no quantitative evaluation, which weakens the generalization claims.
>
> Response: We thank the reviewer for the suggestion. We address the concerns regarding task generalization and real-world evaluation as follows:
>
> 1. Task Generation:
> To illustrate task generalization, we have applied our method to video segmentation via zeroshot transferring. Particularly, we evaluated our approach on the FISHBOWL and MOVID-A datasets and compared it against established methods such as C2F‑Seg and SaVos as follows. Those results included in the revised manuscript as in section 4.5. Comparison with Video Amodal Completion methods. Despite not being directly trained on these benchmarks, our method achieves competitive performance with state-of-the-art approaches that are supervised on these datasets.
>
> | Method  | Setting  | FISHBOWL \$m\text{IoU}\$ | FISHBOWL \$m\text{IoU}\_{\text{occ}}\$ | MOVID‑A \$m\text{IoU}\$ | MOVID‑A \$m\text{IoU}\_{\text{occ}}\$ |
> | ------- | - | :----------------------: | :------------------------------------: | :---------------------: | :-----------------------------------: |
> | SaVOS   | Self-supervised  |           88.63          |                  71.55                 |          60.61          |                 22.64                 |
> | C2F‑Seg | Supervised  |           91.68          |                  81.21                 |          71.67          |                 36.13                 |
> | Ours    | Zeroshot  |           90.34          |                  78.22                 |          68.12          |                 32.74                 |
>
>
>
>
> 2. Real-World Evaluation:
> The objects in our quantitative experiments are real-world objects; only the occlusions are synthetically added. This allows for precise control and access to ground-truth amodal completion RGB content, which are currently impractical to obtain for in-the-wild videos with real occlusions. We acknowledge that this limits full real-world evaluation. However, as constructing reliable ground-truth amodal completions for naturally occluded scenes remains a major challenge in the field, we believe our setup provides a necessary and informative compromise. We are open and eager to explore any viable benchmarks the reviewer might recommend for natural occlusion settings.

---

> ### Author Response · Authors · 2025-05-21
> **Response to reviewer FHRB - Part 3**
>
> ## 4. Qualitative comparisons are not fully aligned: The paper only compares against image-based methods (PA, Pix2Gestalt). A more informative comparison would also include the frame-level-only variant of the proposed model (before motion training) to fairly assess the contribution of the temporal module. (e.g. in figure 4 & 5).
>
>
> Response: We thank the reviewer's suggestion. In response, we have added into the revised manuscript our frame‑level baseline (i.e., the variant of our model without motion training) alongside PA and Pix2Gestalt in Figures 4 and 5. Figure 4 and 5  jointly illustrate the qualitative advantage of our TGVAC over frame‑level baselines under two occlusion regimes. Each figure displays five evenly‑spaced frames (columns) of a video, while the rows list: the source frames, the visible frames, the amodal completion ground truth (GT), ProgressiveAmodal (PA) results, Pix2gestalt (Pix2ges) results, our ablated variant without motion training, and the full TGVAC. Regarding the easy case, both baselines recover only fragments of the car and exhibit broken contours or missing wheels. Meanwhile, the motion‑free variant predicts plausible completion at each frame, these predictions are not consistent throughout the video. For instance, the completed car in the last frame looks quite different compared to the car in other frames due to large occlusion. By exploiting temporal cues, TGVAC reconstructs the entire vehicle that closely matches the GT and remains temporally coherent across all frames. Regarding the hard case, here more than half of a hiking person is obscured by terrain in several frames. PA and Pix2gestalt fail to predict the missing torso and limbs, producing implausible thin shapes. Meanwhile, our no‑motion variant suffers from inconsistency and faint boundaries. In contrast, TGVAC still infers a complete, well‑shaped human figure (including backpack and limbs) and preserves appearance consistency as the person moves. Together, these examples highlight that integrating motion information enables TGVAC to deliver more complete and temporally stable amodal completions, with especially pronounced gains when occlusion is severe.

---

### Review · Reviewer_DB2i · 2025-05-07

**Summary Of Contributions:**

This paper proposes a new synthetic dataset and a new model for the video amodal completion task.
The synthetic dataset is constructed by overlaying another foreground object to create occlusion. Meanwhile, an MLLM is adopted to generate a textual description of the original video as the generation condition.
The new model is a diffusion model, which is fine-tuned from an image diffusion model. The proposed method first fine-tunes the model to generate a completed foreground object for each frame and then adds cross-frame attention to generate a video with a completed foreground object.
Experiments on the constructed new dataset demonstrate improved performance than the previous image amodal completion methods, Pix2Gestalt (CVPR'24) and ProgressiveCompletion (CVPR'24).

**Audience:**

Yes

**Claims And Evidence:**

Yes

**Requested Changes:**

1. A more reasonable way is to directly use a pretrained video diffusion model like CogVideoX or Wan for the completion task.
2. No videos are provided, which makes it hard to evaluate the quality of the videos.
3. A minor problem: citation format. \citep should be used in most cases instead of \cite.

**Strengths And Weaknesses:**

Strength:
1. The main strength is that the paper studies the problem of amodal completion for videos, while previous works mainly focus on images.
2. The paper is well written with a clear structure.
3. The proposed method is straightforward and reasonable.

Weakness:
1. The contribution seems to be incremental. The main contribution is to generalize the amodal completion from images to videos. However, this seems to be a trivial extension. The construction of the synthetic dataset is similar to amodal completion image datasets in Pix2Gestalt. The training of the diffusion model follows a similar way, and the difference is that the video generation requires cross-frame attention to add cross-frame consistency. All these are straightforward extensions of the image amodal completion method to videos.
2. If the method part does not involve too many contributions, it would be better to include more systematic analysis on the model choice and design. A more reasonable way is to adopt pretrained video diffusion models for the task instead of turning an image generation model for the task.

---

> ### Author Response · Authors · 2025-05-21
> **Response to reviewer DB2i**
>
> We are grateful to the reviewer for your constructive feedback. We would like to address the reviewer’s concerns as follow:
>
> ## 1. The contribution seems to be incremental. The main contribution is to generalize the amodal completion from images to videos. However, this seems to be a trivial extension. The construction of the synthetic dataset is similar to amodal completion image datasets in Pix2Gestalt. The training of the diffusion model follows a similar way, and the difference is that the video generation requires cross-frame attention to add cross-frame consistency. All these are straightforward extensions of the image amodal completion method to videos.
>
> Response: We appreciate the reviewer's concerns. Our intention in this work is not only to extend amodal completion from images to videos, but also to establish a simple, unified, and extensible framework that can serve as a baseline and benchmark for future research in this underexplored area. While our model builds upon established image-based techniques, we introduce several non-trivial and novel components to address the unique challenges of the video setting:
>
>
> 1. Dataset:
> a. Text-Guided Completion Dataset: Our dataset is also annotated with natural language captions, which we believe can foster further multi-modal research in this area.
> b. Dataset Design: While inspired by Pix2Gestalt, our video-centric dataset introduces occlusions that vary over time, with a focus on object motion, temporal dynamics, and scene variation, that are absent in static image datasets.
>
> 2. Methodology:
> a. Text-Guided Completion: We integrate text prompts as part of the generation process, enabling the model to produce semantically meaningful completions grounded in language.
> b. Motion-Aware Training: We propose a motion training strategy that explicitly learns temporal consistency across frames.
> c. Temporal Diffusion Module: For inference over longer sequences, we incorporate a temporal diffusion module that enables consistent, frame-aware completion across time.
>
>
> ## 2. If the method part does not involve too many contributions, it would be better to include more systematic analysis on the model choice and design. A more reasonable way is to adopt pretrained video diffusion models for the task instead of turning an image generation model for the task. A more reasonable way is to directly use a pretrained video diffusion model like CogVideoX or Wan for the completion task.
>
> Response: We appreciate the reviewer’s suggestion regarding the use of pretrained video diffusion models. In response, we conducted additional experiments adopting pretrained video diffusion models for the task. As suggested from the reviewer, we leverage CogVideoX. Since CogVideoX is originally designed to generate videos from text alone, we modified it to handle video amodal completion by incorporating additional inputs: the input video and its corresponding visible-region mask. We encoded the video and mask features and concatenated them with the noise vector used during diffusion, allowing the model to condition on both spatial and temporal context. We refer to this modified version as CogVideoX-Amodal. The table below shows the comparison between our TGVAC and CogVideoX-Amodal.
>
>
>
> | Method  | CLIP ↑ | LPIPS ↓ | TC ↑ |
> | ------------- |:-------------:|:-------------:|:-------------:|
> | CogVideoX-Amodal    | 0.87     |0.16| 0.89|
> | TGVAC (Ours)    | 0.92 |0.08 | 0.94|
>
> While pretrained video diffusion models like CogVideoX are powerful, they are not optimized for amodal video completion tasks like ours. As shown in the aforementioned table, our TGVAC achieves better performance across metrics. This is due to the nature of the amodal completion task: unlike general video synthesis from text, amodal completion requires precise preservation of visible content and plausible generation of occluded regions. Our TGVAC is trained with these constraints in mind, leveraging image-based amodal completion knowledge and motion modeling to maintain visual fidelity and temporal coherence.
>
> ## 4. No videos are provided, which makes it hard to evaluate the quality of the videos.
>
> Response: We thank the reviewer’s suggestion. In this revision round, we have uploaded several videos of TGVAC on natural videos. Please refer to the supplementary materials for these videos.
>
> ## 5. A minor problem: citation format. \citep should be used in most cases instead of \cite.
>
> Response:  We appreciate the reviewer’s suggestion. In response, we have updated our revised manuscript with \citep.

---

### Decision · Action_Editor_jPJq · 2025-06-24

**Recommendation:** Reject

**Additional Comments:**

I would recommend the authors to resubmit the paper after a major revision. As an extension of amodal completion to the video domain, the paper needs to present a clearer technical contribution over existing image amodal completion such as Pix2Gestalt, particularly in terms of dataset construction and model design. Moreover, it is important to experimentally validate the effectiveness of the method on video datasets with significant motion dynamics, and to provide stronger empirical evidence for the benefit of the text conditioning component.

As a minor point, Equations (2) and (4) contain several typos and undefined notations that require clarification.

**Audience:**

No

**Audience Explanation:**

The video amodal completion task itself can attract some interest from the TMLR's audience, particularly from the perspective of applications such as video editing. However, as the proposed method closely resembles existing approaches (Pix2Gestalt and diffusion-based video generation) and the experimental evaluation remains limited, I am concerned that it may be difficult for this work to attract interest from the TMLR audience. To gain more interest from the community, it would be necessary to clarify challenges specific to a video amodal completion problem and to propose and validate corresponding solutions.

**Claims And Evidence:**

No

**Claims Explanation:**

This paper proposes a novel synthetic dataset and a diffusion-based model for text-guided video amodal completion. In specific, the method performs sequential frame-level training and motion training using a conditional latent diffusion model originally designed for video generation, conditioned on the textual description of the occluded object and the visible mask from the first frame. The occlusion videos synthesized using the proposed approach are employed during the second motion training phase and are also used for experimental evaluation. Experimental results demonstrate that the proposed model performs competitively as a first baseline model on the video amodal completion task.

Overall, the proposed method seems to be technically sound. However, while this paper builds upon the existing image amodal completion method, Pix2Gestalt, by extending it to handle video data in both data generation and modeling, the benefits in using the text-conditioning as well as the extension of object completion to long videos are not sufficiently supported by the experimental results. Thus, the claims are not sufficiently supported by the evidence provided.

**Resubmission Of Major Revision:**

The authors may consider submitting a major revision at a later time.